# Factors Affecting Adoption and Intensity of Use of Tef-*Acacia decurrens*-Charcoal Production Agroforestry System in Northwestern Ethiopia

**Miftha Beshir** [1,2,*] **, Menfese Tadesse** [2] **, Fantaw Yimer** [2] **and Nicolas Brüggemann** [3,*]

1   College of Agriculture and Environmental Science, Arsi University, Asela P.O. Box 193, Ethiopia
2   Wondo Genet College of Forestry and Natural Resources, Hawassa University,
    Shashemene P.O. Box 128, Ethiopia; menfesetadesse@gmail.com (M.T.); fantawyimer2003@gmail.com (F.Y.)
3   Forschungszentrum Jülich GmbH, Institute of Bio- and Geosciences-Agrosphere (IBG-3, Agrosphere),
    52425 Jülich, Germany
*   Correspondence: mifthabeshirb@yahoo.com (M.B.); n.brueggemann@fz-juelich.de (N.B.)

**Abstract:** The tef-*Acacia decurrens*-charcoal production agroforestry system (TACPA system) is a conventionally and uniquely adopted indigenous potential climate-smart agricultural technology (CSAT) in northwest Ethiopia. This study investigates factors determining farmers' adoption and intensity of use of the TACPA system using a descriptive statistic and a double-hurdle model. A total of 326 farming household heads were selected using multistage random sampling from two purposively chosen provinces. The descriptive statistics showed that 64.42% of the local farmers adopted the TACPA system, and the area covered by the adopter was 0.38 ha. Empirical estimates of the first hurdle revealed that credit, plot ownership, association, primary road distance, asset, farming experience, labor, family size, livestock, tenure, and marginal land influenced the adoption of the TACPA system. On the other hand, estimates of the second hurdle showed that the intensity of use of the TACPA system was determined by age, plot ownership, nativity, primary road distance use, livestock, tenure, secondary road distance, and experience. The complementarity between the adoption of the TACPA system and its intensity of use suggests the necessity of joint socio-economic policies to meet the priority needs of smallholder farmers of the study area and to disseminate the innovation to other parts of Africa with similar environmental conditions.

**Keywords:** climate-smart agriculture; charcoal production; double-hurdle model; food security

## 1. Introduction

Climate-smart agriculture (CSA) is being promoted as a strategy for increasing agricultural productivity and farm income, enhancing adaptation and resilience to climate change, and reducing greenhouse gas (GHG) emissions from agriculture [1]. In Sub-Saharan Africa (SSA), low agricultural productivity is characteristic of smallholder farms due to constraints such as irregular rains, deforestation, land degradation, low soil fertility, climate change, and other problems to further reduce agricultural productivity in the region [2]. The high level of poverty in countries of the region hampers national government efforts to improve resilience to climate change [3]. Low crop yields caused by periodic droughts also have left many smallholder farmers in Ethiopia and elsewhere in Sub-Saharan Africa exposed to food insecurity [4]. For example, out of the almost 425 million people in SSA who are expected to be food insecure in 2020, 35 million are food insecure only as a result of COVID-19's impact on GDP [5].

Smallholder farmers in SSA have developed a number of alternative climate-smart technologies (CSATs) that they can utilize either individually or in combination to build resilience against climate-induced calamities such as severe drought [6]. Agroforestry is one of the paradigmatic alternatives to CSA [6]. In general, it is defined as a land-use

system and practice in which woody perennials are intentionally integrated with crops and/or animals on the same land-management unit [7]. It is one of the widely preferred land management practices for enhancing soil carbon (C) storage and perhaps reducing GHG emissions [8], and improving soil fertility [9]. It is increasingly promoted as part of CSA and broader initiatives to achieve the United Nations Sustainable Development Goals [10].

Smallholder farmers may combine or modify different CSATs with other technologies and practices to address their specific strategies and conditions due to differences in awareness, cultures, objectives, preferences, resource endowments, and socio-economic backgrounds [11]. The CSATs considered in this study, the tef-*Acacia decurrens*-charcoal production agroforestry system (hereafter referred to as the TACPA system), is a conventionally and uniquely adopted and modified indigenous agroforestry system in northwestern Ethiopia [12]. It constitutes tef-*Acacia decurrens* intercropping and a charcoal production from *A. decurrens* tree on the same farmland. It can be counted [13] among the CSATs in Ethiopia, such as crop diversification, mulching, crop rotation, intercropping, conservation agriculture, no-tillage, integrated soil fertility management, improved grazing, and improved water management [13]. The production of charcoal on cultivated land and the application of charcoal (biochar) to the soil of the TACPA system are strategies for reducing greenhouse gas (GHG) emissions, sequestering carbon [14,15], and improving soil fertility [16–18].

The TACPA system has been adopted by the local farmers in the study area for the last thirty years; it is among the two agricultural systems in our study area, along with the tef monocropping system (TM system) from which the TACPA system was changed. The main motivation to establish the TACPA system is to generate additional income from charcoal production and enhance soil properties through the introduction of charcoal debris and leaves of *Acacia decurrens* trees [19,20]. Farmers appraise the establishment of the TACPA system as an income-generating strategy to compensate for the decreasing income from annual crops [21,22]. The part of *Acacia decurrens* not suitable for charcoal making is used by the local farmers as fuelwood [21]. According to recent studies, the TACPA system also has significant advantages in terms of environmental sustainability [19–21].

Diverse factors could motivate smallholder farmers to adopt CSATs including the TACPA system, mainly demography [23,24], farm characteristics [21,25], human capital [11,26], social capital [11,27], and financial capital [2,28]. Other factors, such as farmer support services [26,29] and perception [21,25], could also motivate smallholder farmers to adopt specific CSATs. Unfortunately, among the few studies that have sought to identify the factors of farmer adoption of CSATs singly or in combination [22,23,25], none of them have been on the TACPA system, except that of [22] who investigated factors influencing adoption of TACPA system together with many other sustainable management technologies, and tree density and land allocation decision of *Acacia decurrens*-based taungya system [22] in our study area. In addition, adoption of the TACPA system among stallholders is low in the study area, and adapting it to household, farm, environmental and social conditions also seems to be a challenge [22,30,31].

The findings of this study can support policymakers in developing programs and plans for expanding and intensifying the use of the TACPA system in areas with similar agronomic, ecological, and socio-economic conditions in Africa. Hence, this study is aimed at assessing the adoption and the intensity of use of the TACPA system in order to draw important policy implications for future intervention. The hypothesis of this study is that socio-economic factors explaining the adoption and intensity of use of the TACPA system in northwestern Ethiopia are different.

## 2. Research Methodology

### 2.1. Description of the Study Area

The study sample was taken at random from two purposively selected districts in the Amhara region, namely Fagita Lekoma (10°57′–11°11′ N, 36°40′–37°05′ E) and Dangila (11°16′–36°50′ N, 11°16′–36°50′ E), situated between 2137 and 2350 m a.s.l (Figure 1).

The two districts have together a total population of 173,552 and are parts of the moist subtropical agro-ecological zone in the northwestern highlands of Ethiopia. The mean annual rainfall of the area is 1328 mm, and the average annual temperature of 17.5 °C [32]. A large proportion of rural households in these districts are involved in mixed subsistence crop-livestock agricultural systems. Tef (*Eragrostis tef* Zucc.), barley (*Hordeum vulgare* L.), wheat (*Triticum aestivum* L.), and potato (*Solanum tuberosum* L.) are the main crops. The predominant soil types are Nitisol and Acrisol, and of moderately acidic pH [33]. The terrain is undulating and rugged [34]. The rationale for choosing the region is that local farmers have been investing large resources for the expansion of TACPA system (Figure 2a–d), and the local farmers are changing the tef monocropping system (TM system) (Figure 2d), an indigenous agricultural system dominant in the study area, to TACPA system.

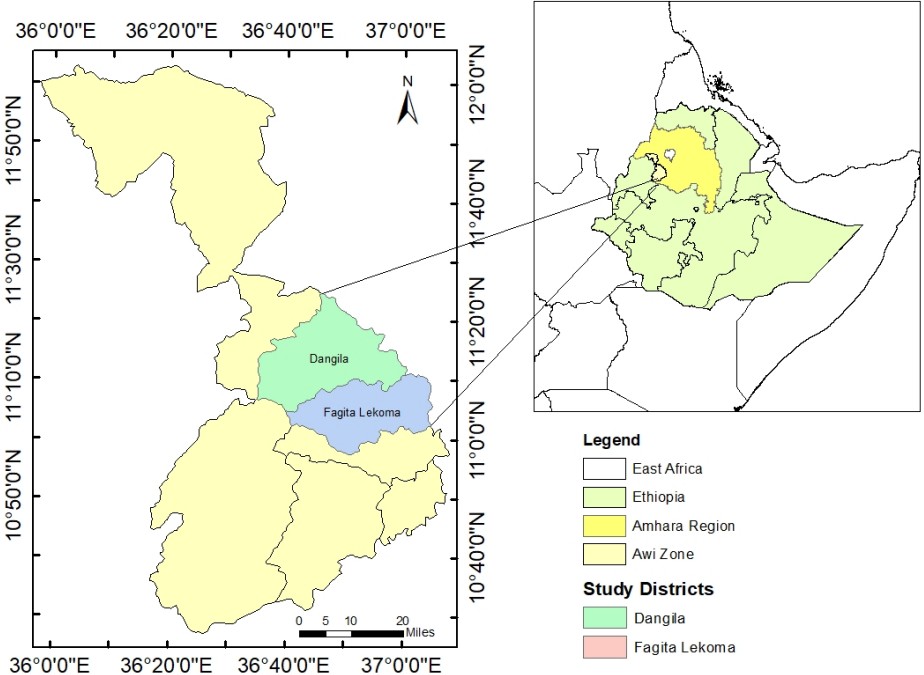

**Figure 1.** Map of the study area.

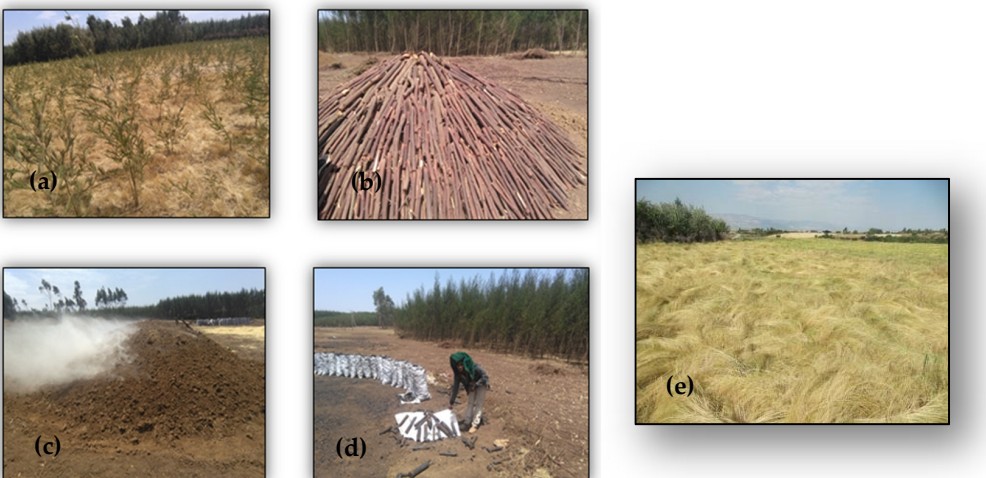

**Figure 2.** The tef-*Acacia decurrens*-charcoal production agroforestry system (TACPA system) in the study area [(**a**) Intercropping of *Acacia decurrens* tree with tef crop (**b**) Pile of Acacia wood ready for charcoal production (**c**) Charcoal production (**d**) Charcoal harvesting], and (**e**) The tef monocropping system (TM system) from which the TACA system developed (Photo by Miftha 2020).

The TACPA system has been practiced in the study area for the last 30 years. Tef (*Eragrostis tef*), an important component of the system, is one of the staple food crops in Ethiopia, covering 2.73 ha of land with average production of 1.28 t ha$^{-1}$ yr$^{-1}$ in Ethiopia as a whole and 1.00 million ha with average production of 1.3 t ha$^{-1}$ the Amhara Region [33]. In the TACPA system, *A. decurrens* trees are planted in the fields with about 25–50 cm spacing immediately after tef are cultivated. The tef is harvested after three to five months, whereas *A. decurrens* is grown for about four to five years. Thereafter, the trees are harvested to produce charcoal in the same field. In the TACPA system, local farmers produce charcoal using mound kilns, which is the oldest method of charcoal making. In the TACPA system, the charcoal residues from under the kiln are spread all over the parcel of the land for soil amelioration.

### 2.2. Sampling Procedure and Data Collection

Prior to the formal survey, district experts, farmers, and development agents were interviewed, and field observation was conducted to gather information for the main survey. A multistage random sampling procedure was used to select respondents. A list of farmers from the two districts was obtained from Fagita Lekoma and Dangila Department of Agriculture Offices in the Amhara region. First, six villages were selected from the two districts based on the expansion and intensity of the TACPA system. Second, 326 farmers were chosen at random from each district and categorized into adopters and non-adopters of the TACPA system. The main survey data were collected during the 2020 cropping seasons using a structured questionnaire after farmers, district experts, and development agents were interviewed, and field observation was conducted to acquire information for the main survey. In order to ensure validity and reliability, the questionnaire was pretested and revised based on feedback and administered by trained enumerators. The questionnaire included valuable information on variety of variables, including farm-related, socio-economic, technological, and institutional factors. Data collected were analyzed using descriptive statistics and double-hurdle model.

### 2.3. Conceptual and Empirical Model

The study concentrated on smallholder farmers' involvement in the establishment of the TACPA system and was designed as a technology adoption study. Establishment, in this case, can be considered as the degree of involvement in the establishment. This can be quantified in terms of total land assigned for the TACPA system. Hence, the selection depends on the maximum use of the TACPA system provided to the adopters and the incentive created by envelopment in its establishment. Adoption continues only when the incentives dominate the disincentives, that is, when the returns are higher than the total costs. However, a variety of factors influence technological adoption. Therefore, distinguishing those factors that hamper adoption is crucial. This is accomplished using several theoretical frameworks. For example, Cragg et al., 1971 [35] underlined that inclination to adopt an innovation is contingent on how a decision-maker acts in the face of a set of constraints and alternatives. In this study, constraints and alternatives are supposed to be dissimilar factors that may influence the smallholder farmers' choices.

Table 1 presents variables such as household head demographics (age, gender, household size, marital status, and nativity), farm characteristics (labor availability, road distance, fragmentation of agricultural plot, availability of marginal land), human capital (farming experience, level of education,), social capital (membership in farmers association, NGO support, training), financial capital (participation in off-farm activities, livestock ownership, household asset, total farm income, and landholding size) farmer support services (access to extension and access to credit) and perception (farmers perception of soil fertility depletion) that were included in the model and their description.

**Table 1.** Description of selected variables that influences the adoption and intensity of use of the tef-*Acacia decurrens*-charcoal production agroforestry system (TACPA system).

| Acronym | Description | Type of Measure | $H_0$ Sign |
|---|---|---|---|
| Dependent variables | | | |
| Adoption | Probability of adoption | Discrete (1 if yes, 0 if otherwise) | |
| Intensity of Adoption | Area covered by TACPA-system | | |
| Explanatory variables | | | |
| Age | Age of the household head | Years | ± |
| Gender | Gender of the household head | 1 if male 0 if female | ± |
| Education | Educational level of the household head | Years | ± |
| Extension | Extension service per year | No. of contacts | + |
| Training | Household received TACPA-system related training | No. of training | + |
| Family size | Family size | Number | + |
| Off-farm activity | Participation in off-farm activities | 1 if yes, 0 if otherwise | ++ |
| Credit | Household access to credit | 1 if yes, 0 if otherwise | + |
| Livestock | Livestock size owned by the household | TLU, tropical livestock unit | + |
| Land holding | Land holding size | In ha | + |
| Plot ownership | Plot ownership | 1 = own, 2 = family, 3 = shared, 4 = rent | - |
| Association | Membership in farmers association | 1 if yes, 0 if otherwise | + |
| Nativity | Nativity | 1 if Native, 0 if non-native | + |
| Tenure | Perception of tenure security | "1" for if a household head feels "secured" and "0" otherwise1 if positive, 0 if otherwise | + |
| Primary road distance | Distance of plot from primary road | Minutes of walking | - |
| Secondary road distance | Distance of plot from secondary road | Minutes of walking | + |
| NGO | NGO support | 1 if yes, 0 if otherwise | + |
| Soil fertility status | Farmers report of soil fertility depletion | 1 = low, 2 = moderate, 3 = high | - |
| Asset | Household asset owned | In ETB (ETB, The Ethiopian Birr, the national currency of the Federal Democratic Republic of Ethiopia) | + |
| Total income | Total farm income | In ETB | + |
| Marginal land | Availability of marginal land | In ha | + |
| Fragmentation | Fragmentation of the agricultural plot | No. | + |
| Experience | Farmers farming experience | Years | + |
| Marital status | Marital status | 1 if married, 0 if otherwise | + |
| Labor | Labor availability in the household | Adult equivalent | + |

*2.4. The Double-Hurdle Model*

The double-hurdle model was used to investigate factors that affected smallholder participation in establishment of the TACPA system and the size of land apportioned to it. The double-hurdle model, which was first proposed by [36], is planned to deal with survey data having a large number of zero observations on a continuous dependent variable [37]. Zeros could be either abstention as in the selection model or corner solutions as in a Tobit model [38]. The double-hurdle model is comparable to the Heckman model in that two sets of parameters are yielded in both cases. Heckman's model, however, has the disadvantage of producing a less efficient estimator than performing less effectively when normality assumption is transgressed [39].

The double-hurdle model has been extensively employed in the consumers' adoption literature [39]. The model supposes that farmers make two independent decisions: whether to adopt or not and the decision on how much to adopt, and those decisions are affected by different variables. Although it has been used to investigate nonfarm decisions in Africa [40], it has not been applied widely to assess socio-economic factors affecting labor allocation in agroforestry practices incorporating charcoal production.

A different latent variable is employed for each decision process in the double-hurdle model. A Probit model is employed to assess the likelihood of participating in the estab-

lishment of the TACPA system by a household, while the truncated regression model is employed to know the intensity of adoption.

The decision (D) to take part in the establishment of TACPA system:

$$D_i = 1 \ldots \text{ if } \ldots D_i^* > 0 \ldots \text{ and}$$

$$D_i = 0 \ldots \text{ if } \ldots D_i^* \leq 0 \tag{1}$$

$$D_i^* = \alpha' Z_i + U_i$$

where, $D_i^*$ is a latent variable that takes the value 1 if a farmer adopts the TACPA system and zero otherwise. $\alpha$ is a vector of parameters and Z is a vector of household characteristics, and $U_i$ is a constant.

The level of participation (Y) after participation decision

$$Y_i = Y_i^* \ldots \text{ if } \ldots Y_i^* > 0 \ldots \text{ and } \ldots D_i^* > 0$$

$$Y_i = 0 \ldots \text{ Otherwise} \tag{2}$$

$$Y_i^* = \beta' X_i + V_i$$

where $Y_i$ is the latent variable relating to the use intensity of adoption (an area of farmland devoted to the TACPA system), $X_i$ is a vector of household socio-economic characteristics, $\beta$ is a vector of parameters, and $V_i$ is a constant.

Allowing for a non-normal error structure and heteroscedasticity [41], the empirical model is determined using maximum likelihood of the form:

$$\text{LogL} = \sum ln\left[1 - \Phi\left(\alpha z_i'\left(\frac{\beta x_i}{\sigma}\right)\right)\right] + \sum ln\left[\Phi(\alpha z_i')\frac{1}{\sigma}\varnothing\left(\frac{Y_i - \beta x_i'}{\sigma}\right)\right] \tag{3}$$

where, $\varnothing$ and $\Phi$ are the cumulative distribution function (cdf) of the standard normally distributed random variable and the probability density function (pdf).

The double-hurdle model was chosen in this study because factors influencing the decision to establish the TACPA system may differ from those influencing the level of intensification. In this scenario, the Tobit model has the drawback of being inseparable from decision of participation and from the proportion of land allocated to the TACPA system. The main issue is how independent variables in the model can affect the establishment decision as they affect the size of land allocated to the TACPA system. Therefore, the study employed the double-hurdle model.

## 3. Results and Discussions

### 3.1. Descriptive Statistics

The *t*-test for the mean of continuous explanatory variables for the sample households is shown in Table 2. The data contain 326 farm households. Of these, about 64.42% are adopters and 35.58% are non-adopters of the TACPA system during the 2019 and 2020 cropping seasons. The area covered by the TACPA system is about 0.38 ha for adopters. Primary road distance, secondary road distance, asset owned, and labor availability all show significant differences between adopters and non-adopters. This shows the presence of a possible association between household success in the TACPA system adoption and these variables (Table 2). Moreover, relatively more adopters were located further from the primary and secondary roads than non-adopters. Similarly, adopters of the TACPA system are endowed with more assets and have more labor than non-adopters. In general, larger mean values suggest that these factors have an impact on the TACPA system's adoption rate. (Table 2).

**Table 2.** Descriptive statistics of continuous variables used to characterize adopters and non-adopters in the study (n = 326), with significance test for the differences between adopters and non-adopters (* Significant at 10%, ** at 5% and *** at 1% probability level).

| Variables | Adopters | | Non-Adopters | | |
|---|---|---|---|---|---|
| | N | Mean | N | Mean | *t*-Statistics |
| Age | 210 | 48.47 | 116 | 48 | 0.8713 |
| Education | 210 | 2.75 | 116 | 2.94 | 0.6389 |
| Extension | 210 | 7.99 | 116 | 7.24 | 0.4217 |
| Training | 210 | 1.63 | 116 | 1.68 | 0.3888 |
| Family size | 210 | 5.27 | 116 | 5.51 | 0.3258 |
| Livestock | 210 | 5.29 | 116 | 4.88 | 0.2191 |
| Land holding | 210 | 1.35 | 116 | 1.27 | 0.3776 |
| Primary road distance | 210 | 17.75 | 116 | 12.69 | 0.0012 * |
| Secondary road distance | 210 | 7.12 | 116 | 4.92 | 0.0027 * |
| Asset | 210 | 78,285.71 | 116 | 63,104.31 | 0.0624 *** |
| Total income | 210 | 33,505.24 | 116 | 29,650.86 | 0.1746 |
| Marginal land | 210 | 0.011 | 116 | 0.015 | 0.6738 |
| Fragmentation | 210 | 3.21 | 116 | 3.21 | 0.9963 |
| Experience | 210 | 27.91 | 116 | 27.89 | 0.9635 |
| Labor | 210 | 3.81 | 116 | 3.44 | 0.0466 ** |

Table 3 shows the chi-square test for comparing the proportion of categorical variables by adoption status and the chi-square test for determining the significant association between categorical explanatory variables of households. Differences in proportions of categorical variables between adopters and non-adopters are observed (Table 3). For example, the proportions of household heads who received credit are higher in adopters than non-adopters. Similarly, the proportion of households who cultivate their own land has a higher adoption rate than non-adopters. Moreover, higher proportions of native households were also observed among adopters than non-adopters. A statistically significant relationship was also observed between the probability of the adoption of the TACPA system and the soil fertility status of farms' land (Table 3).

**Table 3.** Descriptive statistics of discrete variables used to characterize adopters and non-adopters in the study (n = 326), with significance test for the differences between percent of respondents out of adopters (n = 210) and non-adopters (n = 116) (* Significant at 10% and *** at 1% probability level).

| Variables | Discrete | Adaptors | Non-Adopters | Pearson's Chi-Square |
|---|---|---|---|---|
| Gender | Male | 162 | 48 | 0.6710 |
| | Female | 48 | 22 | |
| Off-farm activity | Yes | 71 | 33 | 0.9886 |
| | No | 139 | 83 | |
| Credit | Yes | 143 | 89 | 2.7114 * |
| | No | 67 | 27 | |
| Plot ownership | Own | 142 | 108 | |
| | Family | 12 | 8 | |
| | Shared | 11 | 0 | 37.4319 *** |
| | Rent | 45 | 0 | |
| Association | Yes | 197 | 116 | 7.4792 |
| | No | 13 | 0 | |
| Tenure | Positive | 202 | 113 | 0.3430 |
| | Negative | 8 | 3 | |
| NGO | Yes | 1 | 1 | 0.1825 |
| | No | 209 | 215 | |
| Marital status | Married | 109 | 102 | 0.5182 |
| | Not married | 20 | 14 | |
| Residential status | Native | 184 | 112 | 7.1358 *** |
| | Non-native | 26 | 4 | |
| Soil Fertility | Low | 42 | 25 | |
| | Moderate | 148 | 71 | 4.6705 * |
| | High | 20 | 20 | |

### 3.2. Determinants of Adoption Rate and Intensity of the TACPA System

Table 4 shows the empirical results of the double-hurdle model estimations of the parameters determining TACPA system adoption and intensity of use.

**Table 4.** Factors influencing the adoption and the intensity of use of tef-*Acacia decurrens*-charcoal production system (TACPA system) adoption: The double-hurdle results (n = 326) (* Significant at 10%, ** at 5% and *** at 1% probability level).

| Variables | Probit | | | Truncated | | |
|---|---|---|---|---|---|---|
| | Coefficient | Robust SE | Marginal Effect | Coefficient | Robust SE | Marginal Effect ** |
| Age | −0.023 | 0.018 | −0.007 | 0.017 ** | 0.011 | 0.009 |
| Gender | 0.345 | 0.266 | 0.107 | −0.023 | 0.131 | −0.012 |
| Education | 0.009 | 0.030 | 0.003 | −0.002 | 0.015 | −0.001 |
| Extension | 0.019 | 0.015 | 0.006 *** | 0.004 | 0.005 | 0.002 *** |
| Training | 0.217 | 0.233 | 0.067 | 0.104 | 0.113 | 0.057 * |
| Family size | −0.219 *** | 0.064 | −0.068 *** | −0.060 | 0.027 | −0.033 |
| Off-farm activity | 0.032 | 0.213 | 0.010 | 0.178 | 0.133 | 0.097 ** |
| Credit | 0.663 *** | 0.244 | 0.205 | −0.098 | 0.143 | −0.054 |
| Livestock | −0.096 *** | 0.040 | −0.030 ** | −0.065 *** | 0.026 | −0.036 ** |
| Land holding | 0.080 | 0.160 | 0.025 | 0.455 *** | 0.124 | 0.250 |
| Plot ownership | 1.047 *** | 0.138 | 0.324 *** | 0.143 *** | 0.050 | 0.079 ** |
| Association | 4.823 *** | 0.348 | - | −0.260 | 0.227 | −0.143 |
| Nativity | 1.345 | 0.352 | 0.0416 *** | 0.401 *** | 0.247 | 0.221* |
| Tenure | −0.681 *** | 0.544 | −0.210 | −0.782 ** | 0.079 | −0.410 |
| Primary road distance | 0.037 ** | 0.013 | 0.011 *** | 0.018 *** | 0.004 | 0.010 *** |
| Secondary road distance | 0.036 | 0.026 | 0.011 | −0.020 ** | 0.011 | −0.011 ** |
| NGO | 0.154 | 1.068 | 0.048 | −0.055 | 0.125 | −0.030 |
| Soil fertility status | −0.327 | 0.216 | −0.101 | −0.161 | 0.080 | −0.088 |
| Asset | $3.74 \times 10^{-6}$ ** | $1.75 \times 10^{-6}$ | $1.16 \times 10^{-6}$ ** | $-1.37 \times 10^{-7}$ | $4.89 \times 10^{-7}$ | $-7.52 \times 10^{-8}$ *** |
| Total income | - | - | - | - | - | - |
| Marginal land | −3.660 ** | 1.886 | −1.132 | −1.414 | 1.102 | −0.777 |
| Fragmentation | −0.066 | 0.060 | −0.020 | −0.049 | 0.040 | −0.027 * |
| Experience | 0.037 * | 0.020 | 0.012 ** | −0.020 ** | 0.012 | −0.011 ** |
| Marital status | −0.422 | 0.315 | 0.079 | −0.006 | 0.153 | −0.003 |
| Labor | 0.480 *** | 0.092 | 0.148 *** | 0.081 | 0.037 | 0.044 *** |
| Constant | −8.760 | 2.517 | | 0.293 | 0.723 | |
| Test statistics | LR chi$^2$(24) = 131.10 Log likelihood = −139.716 Number of obs = 313 | | | Wald chi$^2$(25) = 66.56 Log likelihood = −49.199 Number of obs = 212 | | |

The value of the Pseudo R$^2$, the log-likelihood, and the LR-Chi$^2$ indicate that the specifications for the two models provide a good fit to the data and that the explanatory variables used in the models collectively explain farmers' decision to adopt and intensify the TACPA system use.

Because of a slight heteroscedasticity issue, the variance was determined using robust standard error estimation. In order to check for multicollinearity problems, contingency coefficients and variance inflation factors (VIF) were computed for categorical and continuous variables. As shown in Table 5, all continuous variables have VIF values less than 10 and therefore have no pressing multicollinearity issues. Similarly, contingency coefficients calculated for categorical variables were less than 0.75 (Table 6). Hence, there is no pressing collinearity issue among the categorical variables.

In the first hurdle, the variables credit, plot ownership, association, primary road distance, asset, farming experience, and labor had a positive influence on the likelihood of adoption of the TACPA system, while family size, livestock, tenure, and marginal land had a negative effect. In the truncated model, the variables age, plot ownership, nativity, and primary road distance had a positive influence on the intensity of use of the TACPA system. Conversely, livestock, tenure, secondary road distance, and experience had a negative effect. In addition, empirical assessment of the first and second hurdle revealed that factors explaining both the adoption and intensity of use of the TACPA system together are plot ownership, farmers' experience, livestock ownership, and tenure security. The marginal

effects of the Probit and truncated model show a change in the adoption of the TACPA system for a unit increase in the independent or decision variables.

**Table 5.** Variance inflation factors (VIF) of the continuous explanatory variables.

| Variables | Collinearity Statistics | |
| --- | --- | --- |
| | VIF | Tolerance |
| Age | 9.22 | 0.1049 |
| Education | 1.30 | 0.7679 |
| Extension | 1.30 | 0.7671 |
| Training | 1.41 | 0.7103 |
| Family size | 2.23 | 0.4488 |
| Livestock | 1.35 | 0.7402 |
| Land holding | 1.69 | 0.5909 |
| Primary road distance | 2.33 | 0.4285 |
| Secondary road distance | 2.28 | 0.4384 |
| Total income | 1.36 | 0.7366 |
| Marginal land | 1.09 | 0.9206 |
| Fragmentation | 1.63 | 0.6112 |
| Experience | 9.53 | 0.1,49 |
| Labor | 2.38 | 0.428 |

**Table 6.** Contingency coefficients for dummy explanatory variables.

| Variables | Gender | Marital Status | Residential Status | Plot Ownership | Off-Farm Activity | NGO | Credit | Association | Soil Fertility | Tenure |
| --- | --- | --- | --- | --- | --- | --- | --- | --- | --- | --- |
| Gender | 1.000 | | | | | | | | | |
| Marital status | 0.384 | 1.000 | | | | | | | | |
| Nativity | 0.195 | 0.065 | 1.000 | | | | | | | |
| Plot ownership | −0.049 | −0.107 | −0.017 | 1.000 | | | | | | |
| Off-farm activity | 0.006 | −0.024 | −0.102 | −0.169 | 1.000 | | | | | |
| NGO | 0.041 | 0.026 | 0.025 | 0.040 | 0.030 | 1.000 | | | | |
| Credit | −0.069 | −0.172 | −0.039 | 0.057 | −0.044 | −0.037 | 1.000 | | | |
| Association | 0.084 | −0.018 | 0.044 | −0.001 | −0.029 | 0.016 | 0.112 | 1.000 | | |
| Soil Fertility | −0.042 | −0.286 | 0.028 | 0.029 | 0.121 | −0.011 | 0.165 | 0.025 | 1.000 | |
| Tenure | 0.026 | 0.270 | 0.176 | 0.096 | −0.200 | 0.014 | 0.106 | 0.049 | −0.212 | 1.000 |

As anticipated, age had a positive and significant influence on the intensity of the TACPA system use. The marginal effect result shows that an increase in the age of the household head by one year would increase the intensity of use by 1%. This implies that older respondents adopted the TACPA system more than young farmers. The justification for this is that older farmers might have accumulated knowledge, experience, and stock of human capital over the years. This result is in line with earlier works by [4] in Malawi and [21] in Kenya, who showed a significant relationship between age with the use of soil and water conservation practices. This finding also agrees with previous studies on agricultural technology adoption [23,29,42,43]. The result, however, is inconsistent with the findings of [44–46], who asserted that younger farmers are more responsive toward newly introduced technologies than older farmers, who tend to be more risk-averse in testing technology.

The predicted coefficient of family size was negative and significant at the 1% level in determining a farmer's decision to adopt the TACPA system implying that households with a large family size are less likely to adopt the technology than households with a smaller family size. An increase in farm size by one person would decrease the adoption of the TACPA system by 7%. This might be an indication that the TACPA system could be established by households having less than five members. Bekele et al., 2003 [47] also found similar results for soil and water conservation structures in the eastern highlands of Ethiopia. They stated that in households with a greater number of mouths to feed, competition emerges for labor between food generating off-farm activities.

Access to credit was positively and significantly associated with higher adoption of the TACPA system. For farmers with better access to credit, the probability of the adoption

of the TACPA system increased by 66%. As the TACPA system can be taken as a long-term investment, financial sources may encourage its adoption by easing farmers' cash constraints. The use of credit is a way of covering the cost of the establishment of the TACPA system and assists in overcoming challenges to adoption, such as liquidity or lack of savings [7,48]. The results are in line with earlier works of [49,50], who found a positive association between access to credit and the use of chemical fertilizers, high-yielding crops varieties, and irrigation pumps.

The variable livestock was significant and, contrary to expectations, negatively related to both the adoption and intensity of use of the TACPA system. This suggests that households owning larger livestock holdings were less likely to adopt the TACPA system than were households with smaller livestock holdings. An increase in the value of livestock by 1 TLU (tropical livestock unit) would reduce the probability of adoption by 3% and intensity by 0.4 ha. This is due to the fact that households with larger livestock holdings might have concentrated more on livestock than charcoal and crop production. Similar findings were reported for soil and water conservation practices by [51,52] for rural farmers in Ethiopia. The results, however, were inconsistent with the previous findings by, for instance, Asfaw et al. 2004 [53] for the adoption of chemical fertilizers.

Landholding was significantly and positively related to the intensity of TACPA system use, implying that the intensity of TACPA system use escalated with landholding. This might be due to the relatively large farm size requirement of the establishment of the TACPA system. This is also because landholding size is usually a major determinant of technology adoption [54]. Landholding size can affect consequently be affected by the other factors determining adoption, such as adoption costs, risk perceptions, credit constraints, human capital, labor requirements, tenure arrangements, and more. That is why empirical adoption literature considers landholding as the most important determinant [55,56]. This result is in line with [57] in Tanzania, where adoption of soil and water conservation technologies is low among farmers with land constraints.

The finding reveals that plot ownership significantly and positively affected the adoption as well as the intensity of the TACPA system use, suggesting that farmers with their own farmland are expected to have a higher probability of adoption and intensity of use of the TACPA system. The TACPA system is a long-term investment that indicates benefits only after several years of the initial investment. As a result, the farmers restrict the TACPA system investments to their own land as opposed to land acquired through sharing and inheritance because the benefits of their investment will accrue to them and not be shared with anyone else [58]. In another climate-smart technology adoption study, Nhemachena et al., 2014 [59] found that the likelihood of using integrated soil fertility management and animal manure was higher for farmers who own land than for sharecropped and rented plots.

Membership in farmers' associations significantly and positively influenced the probability of engagement of farmers in the TACPA system establishment. Membership in social organizations such as farmers' associations has been proven to play an essential role in influencing farmers' attitudes about new agricultural technologies and hence enhancing the adoption of such practices [60,61] because they serve as a forum for obtaining credit, access to information, and other useful inputs [62]. A study by [23,63] indicates that farmers' exposure to various information sources by being a member of cooperatives and farmers' associations is related to their capacity to analyze risks and benefits and to make use of improved agricultural technologies.

Nativity significantly and positively affected only the adoption of the TACPA system. The result showed that native farmers are more likely to adopt the TACPA-system as compared to non-native farmers. Adoption of the TACPA system among native farmers was higher than among non-native farmers by 0.41. Natives have been in the communities for most of their lives; hence, they are more likely to appreciate the benefits of the technology and have more [64] resources and guarantees for the security of resources. This corresponds to [65], who noted a significant positive association between nativity and adoption.

Contrary to expectations, farmer perceptions of tenure security were found to be negatively associated with both adoption decisions and intensity of use of the TACPA system. This means that land tenure security delivers no incentives for investments in land management practices. The finding is in line with [66] for tree planting in the Amhara region of Ethiopia. This may not necessarily imply that tenure security does not affect decision-making but may mean that a dummy variable may not be a sufficient measure of tenure security. Perception is a broad concept and, therefore, may not be adequately captured using a few questions. This result is contrary to that of [11,64,67,68], who found tenure security an important determinant of the adoption of land management practices.

As expected, primary road distance was found to be positively associated with both adoption and intensity of use of the TACPA system, suggesting that farmers who are relatively far from towns devote more land to the TACPA system as compared to those who are relatively close. This result is somewhat contradictory because farmers pay more money to transport the charcoal to the market over larger distances. An alternative explanation could be that the TACPA system in the study area is less costly. Such less costly systems, as a result, would be practiced in more remote areas, as per von Thünen's theory [69]. Secondary road distance negatively influenced land allocation to the TACPA system at a 5% significance level. With every additional spent traveling to the market, adoption intensity decreased by 0.1%. This is probably due to the limited access of transportation to the market through the secondary road. The proximity of farmers to the market is essential for the reduction of transportation costs [70].

Asset showed a positive and significant impact on the adoption of the TACPA system in the first model. As an asset owned by the household head increases by the value of one Birr, the probability of adoption increases by 0.001% (Table 6), implying the role of asset owned in driving adoption of the TACPA system. This is perhaps due to the fact that wealthier families are more likely to be market-based and are willing to forego subsistence farming for charcoal marketing. In this study, household assets were used in addition to livestock ownership as a proxy for liquidity or access to cash. Household assets are generally taken as capital to be used for exchange. They influence the adoption of the TACPA system positively [13,60,71]. Past studies also showed that proxies of farm households' wealth status, such as household assets, had a positive and significant influence on the adoption of other agricultural technologies, such as improved seed varieties [25].

Contrary to expectations, the availability of marginal land had a significant negative effect on the probability of adopting the TACPA system, indicating that household heads having marginal lands are less likely to adopt the TACPA system relative to those who do not have. This shows that the TACPA system is the dominant means of income in the study area, and farmers might allocate their marginal land for other purposes such as grazing. Moreover, the cultivation of marginal lands is inevitable to meet the increasing demand for food in developing countries because of the prevalent shortage of prime agricultural lands in the densely populated regions [72]. Several studies show that improving food production will necessitate the changing of marginal lands to other suitable land management systems [73,74].

Experience with the TACPA system had a positive impact on its adoption but had no significant impact on its intensity of use. An increase in experience of the household head by one year would increase adoption by 1%, while a one-year increase in experience decreases the intensity of use of the TACPA system use by 1%. This implies that those households with more years of experience are more likely to be aware of soil fertility and income-related benefits of the TACPA system. Furthermore, agricultural technologies could become easier with time and exposure [75]. The insignificant impact of the intensity suggests that the more experienced farmers have invested several years of particular practices, so they may not want to bear additional risks by trying new methods. According to available literature, the experience could show a positive or negative influence on adoption. According to [76], inexperienced farmers did not want to abandon their conventional seed varieties and, therefore, had a lower adoption of pearl millet hybrids. Similarly, Thirtle et al. 2003 [77]

found that farming experience encourages the adoption of BT (genetically modified) cotton in South Africa.

The availability of family labor had a significant and positive impact on the adoption of the TACPA system, showing that it is a determining factor for adoption. An increase in labor availability in a household by one person would increase the adoption of the TACPA system by 15%. This is obvious because the TACPA system is a labor-intensive technology. The result is consistent with the findings of [23,78,79], who found that farmers with larger family labor availability are more willing to adopt labor-intensive CSATs. Similar studies on the role of labor were reported by [4]; in their study, they concluded that labor availability increased the level of adoption of conservation practices.

## 4. Conclusions and Recommendations

This study analyzed factors determining the adoption of the TACPA system using a double-hurdle model and 312 sampled households. The study hypothesized factors affecting the decision to adopt the TACPA system might be different from those influencing the intensity of adoption. The findings showed a positive association between the adoption and the intensity of use of the TACPA system; however, we observed some disparities in terms of factors influencing the two decisions. Economic analysis of decision variables revealed that credit, plot ownership, membership in farmers' associations, primary road distance, asset owned, farmers' experience and labor availability, family size, livestock ownership, tenure security, and availability of marginal land are important variables influencing the adoption of the TACPA system. On the contrary, the decision to intensify the optimum use of the TACPA system is influenced by age, plot ownership, nativity to the study and primary road distance, livestock ownership, tenure security, secondary road distance, and farmers' experience. Factors explaining both the adoption and intensity of use of the TACPA system are plot ownership, farmers' experience, livestock ownership, and tenure security. In order to develop a successful adoption and intensity of use of the TACPA system in the study area in particular and Africa in general, these factors have to be taken into account, concentrating first on factors affecting households' decision of adoption. The complementarity between the adoption of the TACPA system and its intensity of use also suggests the necessity of joint socio-economic policies. Finally, policymakers and stakeholders of the environmental and the agricultural sectors are recommended to aim at incentivizing the adoption and intensity of use of the TACPA system and their impact on its transfer to other parts of Africa that have similar socio-economic and environmental conditions.

**Author Contributions:** Conceptualization, M.B. and M.T.; methodology, M.B. and M.T.; software, M.B.; validation, M.T., N.B. and F.Y.; formal analysis, M.B.; investigation, M.B.; resources, N.B.; data curation, M.B.; writing—original draft preparation, M.B.; writing—review and editing, M.B., M.T., N.B. and F.Y.; supervision, M.T., N.B. and F.Y. All authors have read and agreed to the published version of the manuscript.

**Funding:** This research was funded by Hawassa University Wondo Genet College of Forestry and Natural Resources and Forschungszentrum Jülich.

**Institutional Review Board Statement:** Informed consent was obtained from all subjects involved in the study.

**Informed Consent Statement:** Not applicable.

**Data Availability Statement:** The datasets analyzed in this study are available from the corresponding author on reasonable request.

**Acknowledgments:** M.B is grateful to all respondents for their willingness to provide data, and to all enumerators for their field assistance. M.B. also acknowledges funding by the German Academic Exchange Service (DAAD) in the funding programme "Research Grants—Bi-nationally Supervised Doctoral Degrees, 2019/20" (57440919).

**Conflicts of Interest:** The authors declare no conflict of interest.

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
