# Peer review of "Factors Affecting Adoption and Intensity of Use of Tef-Acacia decurrens-Charcoal Production Agroforestry System in Northwestern Ethiopia"

_sustainability, doi:10.3390/su14084751_

Round 1

Reviewer 1 Report

In my opinion, the article is innovative, and provides important knowledge about a system that could improve not only soil fertility, but also suppose a system that reduces emissions and therefore help combat climate change. The article amply identifies and analyzes the variables that could make the technology adopted and its use intensive, or the obstacles that such adoption faces in order to be adopted by farmers in the study area. In this way, the aim is to better understand the reasons that may be influencing the farmers' decisions. I find it very interesting from a sociological point of view and it helps to better understand the reality of the agricultural sector in the study area.
However, there are some issues that I would like to see clarified by the authors, and some suggestions:

1. The introduction talks about different CSATs in Ethiopia, but missing, for example, the use of microbial biostimulants that increase soil fertility. Or no-till techniques. Aren't they used in the study area?. Could be included in the text
2. The introduction discusses the few studies that have attempted to identify the determinants of farmers' adoption of CSAT alone or in combination, none of which have been in the TACPA system. It could go a little deeper and give more detail, in general, about some of them.
3. In Figure 1, I think it would be convenient to include Africa, to be able to place exactly where Ethiopia is.
4. In section 2 Research Methodology, a paragraph should be included detailing a little more the type of questions included in the questionnaire. And explain if any test has been done to test it to know its consistency, before conducting the interviews with the farmers.
5. Why was the study based only on small farmers? Wouldn't it have been interesting to carry out the questionnaire also to company technicians to find out if being a company could be an adoption variable?
6. In section 2.2 where the equations of the model are shown, the meaning of the components Ui, Vi, α', β' is not explained. In general, this part must be carefully reviewed and all the components of the model must be defined.
7. The source is not indicated in the figures and tables of the article.
8. On line 208 it says Table 3, and I think it is a typo and it refers to Table 2. On line 281 it says table 4 but I think it refers to Table 3.
9. The results comment that family labor had a positive impact on the adoption of the TAPCA system. However, it could have been analyzed whether the profitability of selling charcoal exceeds the opportunity cost of using family labor, and whether this could change the preferences of farmers with greater labor availability.
10. Has the influence that the availability of machinery to carry out the work on the farm could have on the adoption of technology not been studied?
11. A fictitious variable could have been simulated that contemplated some type of subsidy or tax benefit by the Public Administration for those farmers who adopt the technology, and see its influence on the system, in order to be able to establish different policies that encourage its use.
12. In the text, references 43, 44 and 66 do not have a defined hyperlink to the references section.
If the authors respond to my questions, I consider that the article would be ready to be published.
Congratulations to the authors for the research study.
Thank you very much.

Author Response

Response to Reviewer 1

Point 1. The introduction talks about different CSATs in Ethiopia, but missing, for example, the use of microbial biostimulants that increase soil fertility. Or no-till techniques. Aren't they used in the study area? Could be included in the text

Response 1. “No-tillage” is not practiced in the study area, nor inoculation of microbial biostimulants to increase soil fertility. But as no-tillage is widely practiced in some parts of Ethiopia, we have included it in the examples of CSAT in the country (Line 59). The two predominant practices or technologies in the study area are the tef-monocropping system (TM system) and the tef-Acacia decurrens-charcoal production agroforestry system (TACPA system), which was developed from TM system. We have mentioned the TM system in the introduction (Lines 65 and 66), the added a description of TM system at end of the first paragraph in materials and methods (Lines 110 and 111), as well as its picture along with figure 1 for further clarity.

Point 2. The introduction discusses the few studies that have attempted to identify the determinants of farmers' adoption of CSAT alone or in combination, none of which have been in the TACPA system. It could go a little deeper and give more detail, in general, about some of them.

Response 2. The TACPA system is a new potential CSAT, unique to the study area. Few literature is available about it to our knowledge, which has been included as per the suggestion (e.g., references: 22; 30 and 31)

Point 3. In Figure 1, I think it would be convenient to include Africa, to be able to place exactly where Ethiopia is.

Response 3. We have modified the map of the study areas so that it shows the position of Ethiopia relative to east Africa (Line 93).

Point 4. In section 2 Research Methodology, a paragraph should be included detailing a little more the type of questions included in the questionnaire. And explain if any test has been done to test it to know its consistency, before conducting the interviews with the farmers.

Response 4. We have indicated at end of the second paragraph of the research methodology section that “The questionnaire was pretested and amended based on the feedback obtained to ensure validity and reliability, and administered by trained enumerators.” (Lines 139; 140 and 141) before conducting the interviews with the farmers.

Point 5. Why was the study based only on small farmers? Wouldn't it have been interesting to carry out the questionnaire also to company technicians to find out if being a company could be an adoption variable?

Response 5. Questionnaires were not distributed to company technicians, since our research interest was local farmers who are changing their TM system to TACA system. Above all, currently, no company is interested in establishment of the TACPA system in the study area.

Point 6. In section 2.2 where the equations of the model are shown, the meaning of the components Ui, Vi, α', β' is not explained. In general, this part must be carefully reviewed and all the components of the model must be defined.

Response 6. We have defined components:

  • Di* as a latent variable that takes the value 1 if a farmer adopts the TACPA system and zero otherwise.
  • Ui   as constant (previously not included, Line 226)
  • Z as a vector of household characteristics and α is a vector of parameters
  • Yi as the latent variable relating to the use intensity of adoption
  • Xi as a vector of household socioeconomic characteristics and β is a vector of parameter,
  • and Vi is a constant. (Previously not included, Line 234)

Point 7. The source is not indicated in the figures and tables of the article.

Response. The source of Figure 2 was cited as Miftha 2020 (Line 127), and references hyperlinked. The sources of all other figures and tables are authors work.

Point 8. On line 208 it says Table 3, and I think it is a typo and it refers to Table 2. On line 281 it says table 4 but I think it refers to Table 3.

Response 8. Yes, thanks for spotting this typo. Table 3 refers to Table 2, and we have changed it accordingly in the text. In contrast, Table 3 is as it is.

Point 9. The results comment that family labor had a positive impact on the adoption of the TAPCA system. However, it could have been analyzed whether the profitability of selling charcoal exceeds the opportunity cost of using family labor, and whether this could change the preferences of farmers with greater labor availability.

Response 9. The economic profitably of charcoal production in the system has been addressed in detail by Nigussie et al., 2019, Economic and financial sustainability of an Acacia decurrens-based Taungya system for farmers in the Upper Blue Nile Basin, Ethiopia Land Use Policy, 90, 104331 {Reference 78}. This article has been included in the discussion (last paragraph, {Line 538}) about the profitability of the TACPA system in relation to family labor.

In our study, we were also able to show that the availability of family labor had significant and positive impact on the adoption of TACPA system (Lines: 534 and 535), hence on charcoal production.

Point 10. Has the influence that the availability of machinery to carry out the work on the farm could have on the adoption of technology not been studied?

Response 10. In the TACPA system all labor is exclusively performed by farmers, except the use of oxen at the initial phase of TACPA system establishment.  Most farmers in the study area are so poor that they cannot agricultural machinery.

Point 11. A fictitious variable could have been simulated that contemplated some type of subsidy or tax benefit by the Public Administration for those farmers who adopt the technology, and see its influence on the system, in order to be able to establish different policies that encourage its use.

Response 11. Regarding your suggestion to simulate fictitious data to study ‘subsidy or tax benefit by the public administration for those farmers who adopt the technology’, we are afraid, we could distort the double hurdle model by creating multicollinearity problem (or create unacceptable variance inflation factors (VIF) and contingency coefficients), and therefore did not follow your recommendation here.

Point 12. In the text, references 43, 44 and 66 do not have a defined hyperlink to the references section.

Response 12. The mentioned hyperlinks have been provided.

Reviewer 2 Report

I carefully analyzed the paper. I consider that the paper addresses an important issue regarding sustainability. However, the paper requires major improvements. I consider that the paper is well documented, but then the methodology and results are not appropriate.
To improve the paper, I recommend the following:
- the abstract must be reconstructed to be a miniature of the paper; this should not be a summary of the introduction; more information from the research methodology and results must be entered.
- the introductory part is appropriate from my point of view.
- the research methodology must clearly describe what the TACPA system represents and what is the role of this system in the sustainability of the area; the components of the TACPA system must be described; the climatic conditions of the area must be described.
- the research methodology must be specified.
- all abbreviations used in tables and figures, from the results, must be explained in footnotes.
- Discussions should be addressed in relation to the results in terms of sustainability of the area.

Author Response

Response to Reviewer 2

Point 1. the abstract must be reconstructed to be a miniature of the paper; this should not be a summary of the introduction; more information from the research methodology and results must be entered.

Response 1. The abstract was modified as suggested: the introduction sentence was shortened; the results were written in detail

Point 2. the introductory part is appropriate from my point of view.

Response 2. The introduction has been modified a little as per the suggestion of reviewer two, and a new paragraph (fourth paragraph) has been included in relation to sustainability of TACPA system. But its initial content and structure are the same

Point 3. the research methodology must clearly describe what the TACPA system represents and what is the role of this system in the sustainability of the area; the components of the TACPA system must be described; the climatic conditions of the area must be described.

Response: The TACPA system has been described in the research methodology (second paragraph added. Lines: 112 to 122), and its suitability elaborated in the fourth paragraph of the introduction (Lines 64 to 73).  To describe the climate, the mean annual rainfall and the average annual temperature of the study area were given along with its type agroecology (Lines 100 to 103).

Point 4. the research methodology must be specified.

Response 4. The research methodology has been specified: a survey method was used (now clearly written in the beginning and in the middle of section 2.2{Lines 129)

Point 5. all abbreviations used in tables and figures, from the results, must be explained in footnotes.

Response 5. All abbreviations in the table and figures were written in full in brackets, when necessary abbreviations were explained in the foot note.

Point 6. Discussions should be addressed in relation to the results in terms of sustainability of the area.

Response 6: The sustainably of the TACPA system has been discussed sufficiently in the fourth paragraph of the introduction section (Lines 64-73); and in the discussion section in one way or another.

Round 2

Reviewer 2 Report

I consider that the authors made all the corrections and improvements requested by me. Currently, the paper is much clearer, and the research methodology is better explained, the paper being better understood.

Author Response

Please see the attachment of the corrected manuscript (duplicate sentences significantly reduced from the manuscript as per the request)
